# Synthesis and Biological Activity of Piperidinothiosemicarbazones Derived from Aminoazinecarbonitriles

**DOI:** 10.3390/ph16091267

**Published:** 2023-09-07

**Authors:** Dagmara Ziembicka, Katarzyna Gobis, Małgorzata Szczesio, Ewa Augustynowicz-Kopeć, Agnieszka Głogowska, Izabela Korona-Głowniak, Krzysztof Bojanowski

**Affiliations:** 1Department of Organic Chemistry, Faculty of Pharmacy, Medical University of Gdańsk, 107 Gen. Hallera Ave., 80-416 Gdansk, Poland; 2Institute of General and Ecological Chemistry, Faculty of Chemistry, Lodz University of Technology, Zeromskiego 116, 90-924 Lodz, Poland; malgorzata.szczesio@p.lodz.pl; 3Department of Microbiology, Institute of Tuberculosis and Pulmonary Diseases, 26 Płocka Str., 01-138 Warsaw, Poland; e.kopec@igichp.edu.pl (E.A.-K.); araceli@op.pl (A.G.); 4Department of Pharmaceutical Microbiology, Faculty of Pharmacy, Medical University of Lublin, 1 Chodźki Street, 20-093 Lublin, Poland; iza.glowniak@umlub.pl; 5Sunny BioDiscovery Inc., 972 East Main Str., Santa Paula, CA 93060, USA; kbojanowski@sunnybiodiscovery.com

**Keywords:** pyridine, pyrazine, thiosemicarbazone, synthesis, tuberculostatic activity, antimicrobial activity, cytotoxic activity, zwitterionic form, ADME, structure–activity relationship

## Abstract

To investigate how structural modifications affect tuberculostatic potency, we synthesized seven new piperidinothiosemicrabazone derivatives **8**–**14**, in which three of them had a pyrazine ring replacing the pyridine ring. Derivatives **8**–**9** and **13**–**14** exhibited significant activity against the standard strain (minimum inhibitory concentration (MIC) 2–4 μg/mL) and even greater activity against the resistant *M. tuberculosis* strain (MIC 0.5–4 μg/mL). Additionally, the effects of compounds **8**–**9** were entirely selective (MIC toward other microorganisms ≥ 1000 μg/mL) and non-toxic (IC50 to HaCaT cells 5.8 to >50 μg/mL). The antimycobacterial activity of pyrazine derivatives **11**–**12** was negligible (MIC 256 to >500 μg/mL), indicating that replacing the aromatic ring was generally not a promising line of research in this case. The zwitterionic structure of compound **11** was determined using X-ray crystallography. Absorption, distribution, metabolism, and excretion (ADME) calculations showed that all compounds, except **11**, could be considered for testing as future drugs. An analysis of the structure–activity relationship was carried out, indicating that the higher basicity of the substituent located at the heteroaromatic ring might be of particular importance for the antituberculous activity of the tested groups of compounds.

## 1. Introduction

*Mycobacterium tuberculosis* is an acid-resistant Gram-positive bacterium that causes tuberculosis [1]. It is estimated that approximately 30% of the global population are carriers of this bacterium, and every year, the World Health Organization reports numerous diseases and deaths caused by its infection [2]. The tuberculosis treatment process is very demanding for patients, not only because of the extended duration of therapy but also because of the side effects that occur after taking tuberculosis drugs, such as nausea, hearing loss, or hepatotoxicity [3,4]. Immunocompromised patients, including those using immunosuppressive drugs or who are HIV-positive, are at higher risk of tuberculosis [5,6]. The HIV (human immunodeficiency viruses) carrier status increases the susceptibility of patients to mycobacterium infection and facilitates the reactivation of latent infection [7]. An additional factor that worsens the epidemic situation in countries where the clinical situation related to tuberculosis is stable is migration and the influx of people from areas where both the diagnosis and treatment of tuberculosis are not prioritized [8]. The COVID-19 pandemic that has impacted the world in recent years has further worsened patients’ access to diagnosis and treatment [9,10]. However, the most alarming aspect is still the problem of the increasing drug resistance of *M. tuberculosis*, which significantly reduces the effectiveness of the therapy used and increases the number of deaths [11,12]. In this situation, it is not surprising that public health authorities are collaborating to develop multidirectional therapeutic methods that would help overcome the tuberculosis problem. These efforts include searching for new antituberculosis drugs (Bedaquiline, Delamanid, Pretomanid, etc.), introducing new substances to existing treatment regimens (e.g., Nix-TB trial), testing the antituberculosis activity of drugs previously registered for other diseases (e.g., Metformin), using host-directed therapy (e.g., Niraparib), or using drug delivery systems (e.g., chitosan) [13,14,15,16].

Isoniazid (INH) and pyrazinamide (PZA) are first-line drugs used in the treatment of tuberculosis. Pyridine and pyrazine contained in their structures are chemical scaffolds with increasing biological importance [17,18]. In recent years, there has been a growing trend toward designing hybrid drugs that combine multiple pharmacophores in a single chemical molecule. Such treatments are expected to be more potent, due to their affinity for multiple molecular targets. Researchers are exploring this approach for substances with potential anticancer [19], antibacterial [20], or antituberculosis [21] properties.

In line with this modern insight to drug design, we attempted to synthesize pyridine derivatives possessing additional nitrogen-based pharmacophore groups in the molecule. Our previous experiments led us to conclude that compounds containing both a pyridine ring and an amidoxime or benzazole group, as well as cyclic amine, exhibit significant antibacterial and tuberculostatic activities [22,23]. Derivatives containing a thiosemicarbazone group with a cyclic amine attached to the pyridine ring in the C-2 position through an amidrazone linker (Figure 1) are particularly effective against *M. tuberculosis* [24]. Amidrazones are also known for their broad-spectrum antimicrobial activities against Gram-positive bacterial strains (e.g., methicillin-resistant *Staphylococcus aureus*, *Bacillus cereus* [25], *Enterococcus faecium*, and *Streptococcus agalactiae* [26]), Gram-negative bacterial strains (e.g., *Salmonella typhimurium* [27], and multi-drug resistant *Escherichia coli* [28]), and fungal strains (e.g., *Candida albicans* and *Candida krusei* [29]).

This study focused on the synthesis and evaluation of the tuberculostatic activity of pyridine and pyrazine derivatives. These derivatives were substituted in the C-6 position of the pyridine ring, with a morpholine, pyrrolidine, or piperidine ring (or in the C-4 position of the pyridine ring with a piperidine ring) and with piperidine at the end of the thiosemicarbazone group. Two analogous compounds (**DMK-20** and **DMK-16** with morpholine and pyrrolidine in the C-4 position, respectively) synthesized previously were included in the analysis of the results [24]. We also investigated the impact of replacing the pyridine ring with a pyrazine ring on tuberculostatic activity (Figure 2). This research project was a continuation of our previous study on 2,6-disubstituted pyridine [30].

## 2. Results and Discussion

### 2.1. Chemistry

The synthetic pathway employed in this research is depicted in Figure 3. The starting compound, 6-chloropicolinonitrile, 4-chloropicolinonitrile, or 6-chloropyrazine-2-carbonitrile, was subjected to reflux, heated at 60 °C, or stirred for 1 h with a nucleophilic agent (morpholine, pyrrolidine, or piperidine) and a DBU (1,8-diazabicyclo [5.4.0]undec-7-ene) in dioxane, resulting in the formation of 6-substituted or 4-substituted nitriles **1**–**7** (yields 67–100%). In the case of 6- and 4-substituted picolinonitriles **1**–**3** and **7**, the next step involved the conversion of nitrile groups into methyl imidate groups. The reactions were carried out in the presence of methanol and DBU at boiling temperature for 4 h. To obtain the final products **8**–**10** and **14** (yields 68–95%), the methyliminoesters formed in situ were condensed at reflux for 0.5–2 h with piperidine-1-carbothiohydrazide. In the case of 6-substituted pyrazine-2-carbonitriles **4**–**6**, the final products **11**–**13** (yield 33–64%) were achieved by direct reaction between piperidine-1-carbothiohydrazide and nitriles for 12 h at room temperature. The synthesis method of piperidine-1-carbothiohydrazide was known [31]. The products were characterized by spectroscopic methods (IR, ^1^H NMR, ^13^C NMR) and elemental analysis. The characterizations of all tested compounds, along with the ^1^H and ^13^C spectra (Appendix A), are included in the Appendix A.

### 2.2. Biological Activities

#### 2.2.1. Tuberculostatic Activity Assay

The results of the tuberculostatic activity toward *M. tuberculosis* strains H_37_Rv and Spec. 210 are presented in Table 1 as minimum inhibitory concentrations (MICs), which represent the lowest compound concentrations that prevent the visible growth of microorganisms. INH was used as the standard medicine. MICs < 520 μg/mL were considered noteworthy. The tested compounds showed a wide range of MIC values, ranging from 0.5 to above 512 μg/mL. Piperidinothiosemicarbazone derivatives of pyridine **9**–**10** and **14** and pyrazine **13** exhibited a very strong inhibitory effect on *M. tuberculosis* growth. Compared with the MIC values assigned to the reference drug (0.125 and 8 µg/mL), compound **14** showed values that were 16 times higher (MIC 2 µg/mL) for the standard strain and, more importantly, two times lower (MIC 4 µg/mL) for the resistant strain. For compound **10**, these values were even more favorable, amounting to 16 times greater (MIC 2 µg/mL) and 16 times lower (MIC 0.5 µg/mL), respectively. Compound **13** inhibited the growth of both strains at a concentration of 4 μg/mL. The antituberculosis activity of compounds **9**–**10** was found to be at the same level as that of the analogous derivatives described previously, containing a pyrrolidine substituent of similar basicity instead of piperidine on the side chain. In the same reference, an explanation for the lower or higher inhibitory titer for the clinical strain compared with that of the reference strain was provided [30]. Pyridine derivatives **8**, **DMK-20**, and **DMK-16** retained equally strong activity (MIC 4–6.25 µg/mL) against the *M. tuberculosis* strain Spec. 210, with a simultaneous decrease in activity (MIC 6.25–16) against strain H_37_Rv. However, for the pyrazine derivatives **11**–**12**, a complete or almost complete loss of tuberculostatic potency (MIC 256 to >512 µg/mL) was observed. Based on the results of the presented studies, certain conclusions can be drawn regarding the relationship between the structural modifications of compounds and their activity, indicating that for 2,6-disubstituted piperidinothiosemicarbazone derivatives considered in this case, replacing the pyridine ring with a pyrazine ring had an adverse effect on their antimycobacterial activity. The exception was the substitution at the C-6 position with the piperidine ring, which, similar to the 4-substituted and 6-substituted pyridine derivatives, was characterized by the highest potency among its group. For the 6-substituted pyridine and pyrazine derivatives, it was observed that the decrease in activity was in the following order: piperidine > pyrrolidine > morpholine. Piperidine and pyrrolidine were more basic substituents than morpholine. Additionally, the pyrazine ring was less basic than the pyridine ring.

#### 2.2.2. Antimicrobial Activity Assay

The results of the antimicrobial activity against Gram-positive bacteria, Gram-negative, and fungi are presented in Table 2 as MICs, with vancomycin, ciprofloxacin, and fluconazole used as standard medicines. MICs < 1000 μg/mL were considered noteworthy. The MIC values of compounds **9** and **10** for all tested strains were ≥1000, which indicated their lack of extended antibacterial and antifungal activities and their complete selectivity toward *M. tuberculosis*. Such an antimicrobial effect could not be achieved for the previously tested pyrrolidinothiosemicarbazone derivatives and compound **14** [30]. Compound **14** showed higher bacteriostatic activity toward all types of Gram-positive bacteria (MIC 0.06–0.12 μg/mL) than against tuberculostatic activity (MIC 2 μg/mL). Moreover, its growth inhibition potential was stronger than those of the reference drugs CIP and/or VAN. For example, the MIC value for **14** for both *Staphylococcus* strains was 0.06 µg/mL, whereas for CIP and VAN, it was 0.49 µg/mL and 0.98 µg/mL, respectively. In contrast, weak to no activity (MIC 250–1000 µg/mL) was observed for Gram-negative bacteria and fungal strains. Compound **13**, despite the lack of activity against most of the tested strains of microorganisms, was characterized by good activity against two strains of Gram-positive bacteria, namely *S. epidermidis* (MIC 15.6 µg/mL) and *M. luteus* (MIC 62.5 µg/mL).

#### 2.2.3. Cytotoxic Activity Assay

Cytotoxicity toward the HaCaT cell line established using the MTT method (determining cell metabolic activity) and the sulforhodamine B method (determining cell viability) is presented in Table 3 as half-maximal inhibitory concentrations (ICs50), which represent the potency of a compound in inhibiting a specific biological/biochemical function. The SI was calculated as the ratio between IC50 values for noncancerous epithelial HaCaT cells and MIC values against the *M. tuberculosis* reference strain. SI values > 1.0 indicated the nontoxicity of compounds **9**–**10**. For example, for derivative **9**, the MIC value for each method was more than 12.5 times lower than the concentration that evokes a cytotoxic effect toward noncancer cells.

### 2.3. X-ray Study

Compound **11** crystallized in the monoclinic crystal system. The details of crystal data, data collection, and structure refinement are summarized in Table 4. The morpholine ring was disordered. Amidrazone derivatives could exist in two tautomeric forms. The compound in the crystal state assumed the zwitterionic form, stabilized by intramolecular hydrogen bonds of the N⋯H(N)⋯S type (Figure 4). This dipolar form and the planar conformation part of the molecule were additionally stabilized by the N4–H⋯N2 intramolecular bond.

The packing of molecules in the crystal was determined by intermolecular N3–H3⋯S1 hydrogen bonds, which formed a pair of infinite C(7) chains [32] related by inversion and parallel to the [001] direction (Figure 5 and Table 5).

### 2.4. ADME Analysis 

For each tested compound, a bioavailability radar was created (Figure 6, with the pink area representing the optimal range for each property, including lipophilicity (XLOGP3 ranged from −0.7 to +5.0), size (MW ranged from 150 to 500 g/mol), polarity (TPSA ranged from 20 to 130 Å^2^), solubility (log S ≤ 6), saturation (fraction of carbons in the sp^3^ hybridization not less than 0.25), and flexibility (≤9 rotatable bonds) [33]. Only the INH and PZA compounds (first-line anti-TB drugs) did not fall within the optimal range on the bioavailability radar. This was because of the fact that the fraction Csp^3^ was too low. The calculations indicated that the tested compounds were predicted to be orally bioavailable. With regard to drug similarity, the compounds had a good bioavailability score (0.55). All synthesized compounds (except inactive **11**, which did not follow only the Ghose rules) followed the rules of Lipinski [34], Ghose [35], Egan [36], Veber [37], and Muegge [38], which suggested that they were suitable candidates for drugs. The logKp values for the tested compounds ranged from −8.20 to −6.56 cm/s, with more negative logKp values indicating lower permeability of the molecule through the skin.

Analyzing the relationship between the activity and the lipophilicity, the compounds were divided into three groups: 2,6-disubstituted pyridine derivatives, 2,6-substituted pyrazine derivatives, and 2,4-disubstituted pyridine derivatives (Table 6). In each of them, it was noted that the stronger the antimycobacterial activity, the higher the log*P* value. Due to the specific structure of the *M. tuberculosis* cell wall, drugs should be appropriately lipophilic in order to increase their permeability and absorption and to strengthen the protein/membrane binding. Nevertheless, too high a lipophilicity entails lower solubility, which, in combination with extensive first-pass metabolism, may lead to low drug bioavailability. The optimal log*P* value for antitubercular medications was in the range of 1.3–4.1, which is consistent with the obtained results [39]. The only compound not meeting this rule, **11**, was not active against *M. tuberculosis*.

The BOILED-Egg plot (Figure 7 indicated that tested compounds showed absorption in the gastrointestinal tract (white area in Figure 7) and no penetration of the blood–brain barrier (yellow area in Figure 7). Additionally, all compounds were found to be actively effluxed by P-glycoprotein, represented as (PGP+), which is indicated by the blue color of the indicator of the compound.

### 2.5. Structure–Activity Relationship

The results presented above allowed us to determine the structure–activity relationship of the tested piperidinothiosemicarbazone derivatives. The influence of the type of aromatic ring (pyridine or pyrazine), the type of functional group (morpholine, pyrrolidine, or piperidine), and the place of connection of the functional group with the aromatic ring (C-6 or C-4) on the biological activity were investigated. The strongest antituberculous activity was shown by compounds **10**, **13**, and **14**, which contained a piperidine group. Therefore, in this case, the type of substituent—with higher basicity and lipophilicity—seemed to be of the greatest importance. Compound **9** with a pyrrolidine moiety attached to the 6-position of the pyridine ring (with a basicity similar to piperidine but with lower lipophilicity) turned out to be equally active. Studies of antimicrobial activity allowed us to select two compounds, **9**–**10**, of the above-mentioned derivatives (with pyrrolidine and piperidine in the C-6 position of the pyridine ring) characterized by complete selectivity toward *M. tuberculosis* strains. Moreover, cytotoxicity tests showed no harmful effects on nontumor cells. In this way, two leading structures were selected. A promising direction of research is the search for candidates for an antituberculous drug among 2,6-disubstituted thiosemicarbazone pyridine derivatives, with substituents of even higher basicity than piperidine and pyrrolidine (e.g., 1-methylpiperazine or some polycyclic amine fragments).

## 3. Materials and Methods

### 3.1. Chemistry

The starting compounds, reagents, and solvents (Sigma-Aldrich, Darmstadt, Germany) used in this study were of analytical reagent grade. Thin-layer chromatography was performed on an aluminum plate precoated with layers of silica gel 60F_254_ (Merck, Darmstadt, Germany). The chromatograms were visualized using UV light. The stationary base in column chromatography (70–230 mesh) was high-purity grade silica gel 60 (Merck, Darmstadt, Germany). Melting points were measured once via a Stuart SMP30 apparatus (Stone, Staffordshire, UK). IR spectra were recorded using KBr pellets on a Satellite FT-IR spectrophotometer (Bruker, Madison, WI, USA). ^1^H and ^13^C NMR spectra in DMSO-*d*_6_ were recorded using a Varian Unity Plus (500 MHz) instrument (Varian Medical Systems, Palo Alto, CA, USA). The elemental analyses (%C, H, and N) were determined using a Perkin Elmer PE 2400 Series II CHNS analyzer (Perkin-Elmer, Shelton, CT, USA). The results of the elemental analyses were consistent with the calculated values within the ±0.4% range.

#### 3.1.1. Procedure for the Preparation of Nitriles **1**–**7**

##### Method A (**1**, **2**, **4**, **6**, **7**)

First, 40 mmol of an appropriate nitrile, 48 mmol of an appropriate nucleophilic agent, and 6 mL of DBU were dissolved in 25 mL of dioxane. The mixture was refluxed (**1**, **2**, **7**), stirred (**4**), or heated at 60 °C (**6**) for 1 h. After the evaporation of the solvent, ice was added, resulting in the formation of a precipitate, which was then filtered and purified by column chromatography and/or recrystallized from a suitable solvent.

##### Method B (**3**, **5**)

First, 40 mmol of an appropriate nitrile, 48 mmol of an appropriate nucleophilic agent, and 6 mL of DBU were dissolved in 25 mL of dioxane. The mixture was refluxed for 1 h. After the evaporation of the solvent, ice was added, resulting in the formation of an oily suspension, which was neutralized with concentrated hydrochloric acid and then extracted with chloroform (3 × 20 mL). The combined organic phases were dried over anhydrous MgSO_4_. The drying agent was filtered off, and the solvent was evaporated. After triple washing with anhydrous diethyl ether, a precipitate formed, which was then purified by column chromatography and/or recrystallized from a suitable solvent.

#### 3.1.2. Procedure for the Preparation of Piperidinothiosemicarbazones **8**–**14, DMK-20**, and **DMK-16**

##### Method A (**8**)

To a solution containing nitrile (2 mmol) in methanol (15 mL), DBU (0.4 mL, 2.7 mmol) was added, and the mixture was heated to reflux for 4 h. Piperidine-1-carbothiohydrazide (0.318 g, 2 mmol) was then added, and the mixture was refluxed for an additional 0.5 h. The reaction mixture was poured onto ice (40 g) and acidified with acetic acid, resulting in the formation of an oily suspension, which was extracted with chloroform (3 × 15 mL). The combined organic layers were dried over anhydrous MgSO_4_. The drying agent was filtered, and the solvent evaporated. After triple washing with diethyl ether, a precipitate was formed. The precipitated product was dried and recrystallized from suitable solvents.

##### Method B (**9**, **DMK-16**)

To a solution containing nitrile (2 mmol) in methanol (15 mL), DBU (0.4 mL, 2.7 mmol) was added, and the mixture was heated to reflux for 4 h. Piperidine-1-carbothiohydrazide (0.318 g, 2 mmol) was then added, and the mixture was refluxed for an additional 0.5–1.5 h. The reaction mixture was poured onto ice (15 g), resulting in the formation of a precipitate. The precipitated products were filtered, dried, and purified by column chromatography.

##### Method C (**10**,**14**, **DMK-20**)

To a solution containing nitrile (2 mmol) in methanol (15 mL), DBU (0.4 mL, 2.7 mmol) was added, and the mixture was heated to reflux for 4 h. Piperidine-1-carbothiohydrazide (0.318 g, 2 mmol) was then added, and the mixture was refluxed for an additional 1–2 h. The reaction mixture was poured onto ice (40 g) and acidified with acetic acid, resulting in the formation of a precipitate. The precipitated products were filtered, dried, and recrystallized from a suitable solvent.

##### Method D (**11**)

The solution containing nitrile (3 mmol) in 5 mL of methanol was subjected to treatment with DBU (0.2 mL, 1.3 mmol). After stirring for 5 min, 15 mL of methanol and 3 mL of water, in which piperidine-1-carbothiohydrazide (0.478 g, 3 mmol) was dissolved, were added. The mixture was then heated to reflux and left overnight on a stirrer at room temperature. Upon pouring the reaction mixture onto 20 g of ice and acidifying with acetic acid, an oily suspension formed, which was extracted with chloroform (3 × 10 mL). The combined organic layers were dried over anhydrous MgSO_4_. The drying agent was filtered, and the solvent evaporated. After triple washing with diethyl ether, a precipitate was formed. The precipitated product was dried and purified by column chromatography and then recrystallized from a suitable solvent.

##### Method E (**12**,**13**)

The solution containing nitrile (3 mmol) in 5 mL of methanol was subjected to treatment with DBU (0.2 mL, 1.3 mmol). After stirring for 5 min, 15 mL of methanol and 3 mL of water, in which piperidine-1-carbothiohydrazide (0.478 g, 3 mmol) was dissolved, were added. The mixture was then heated to reflux and left overnight on a stirrer at room temperature. Upon pouring the reaction mixture onto 20 g of ice (**12**) and acidifying with acetic acid (**13**), a precipitate formed. After triple washing with diethyl ether, a precipitate was formed. The precipitated products were filtered, dried, and recrystallized from a suitable solvent.

### 3.2. Biological Activities

#### 3.2.1. Tuberculostatic Activity Assay

The tuberculostatic activity of the newly synthesized piperidinothiosemicarbazones **8**–**14** was evaluated in vitro using two strains of *M. tuberculosis*: the standard H_37_Rv ATCC25618 (LGC Standards, Teddington, Middlesex, UK) and a strain isolated from tuberculosis patients (National Tuberculosis and Lung Diseases, Warsaw, Poland) Spec. 210, which was resistant to INH, rifampicin, *p*-aminosalicylic acid, and ethambutol. The broth microdilution method described in detail earlier [30] was employed for the investigations, with INH being used as the reference drug for comparison. Each experiment was repeated in triplicate.

#### 3.2.2. Antimicrobial Activity Assay

The antimicrobial activity of compounds **9**–**10** and **13**–**14** was tested in vitro against various strains of Gram-positive bacteria (*Bacillus cereus* ATCC 10876, *Bacillus subtilis* ATCC 6633, *Staphylococcus epidermidis* ATCC 12228, *Staphylococcus aureus* ATCC 25923, and *Micrococcus luteus* ATCC 10240), Gram-negative bacteria (*Pseudomonas aeruginosa* ATCC 9027, *Proteus mirabilis* ATCC 12453, *Escherichia coli* ATCC 25922, and *Klebsiella pneumoniae* ATCC 13883), and fungal strains (*Candida parapsilosis* ATCC 22019 and *Candida albicans* ATCC 1022310) (LGC Standards, Teddington, Middlesex, UK). The investigations were conducted via the microdilution broth method, as described in detail earlier [40]. Vancomycin, ciprofloxacin, and fluconazole were employed as reference drugs for comparison. Each experiment was repeated in triplicate.

#### 3.2.3. Cytotoxic Activity Assay

The cytotoxic activity of compounds **9**–**10** was evaluated in vitro against the HaCaT cell line, which is an immortalized human keratinocyte line (tissue: skin; age: 62, and gender: male) obtained from AddexBio, San Diego, CA, USA. The MTT method and the sulforhodamine B method, described in detail earlier [30], were used for the assays. The colorimetric signal was acquired using a Molecular Devices MAX190 microplate reader and SoftMax3.1.2PRO software. Statistical significance was assessed by conducting two-tailed student’s *t*-tests. Deviations ≥ 20% compared to the water control, with *p*-values less than 0.05, were considered statistically significant. Each experiment was repeated in triplicate.

### 3.3. X-ray Study

Single-crystal X-ray diffraction data were collected at 100 K on an XtaLAB Synergy, Dualflex, Pilatus 300 K diffractometer apparatus (Rigaku Corporation, Tokyo, Japan) for compound **11**. Using Olex2 [41], the structure was solved with the SHELXT [42] structure solution program using Intrinsic Phasing and refined with the SHELXL [43] refinement package using least-squares minimization. The CIF file containing full crystallographic data for compound **11** (CCDC 2278736) can be obtained free of charge from The Cambridge Crystallographic Data Centre via http://www.ccdc.cam.ac.uk/structures (accessed on 11 July 2023).

### 3.4. ADME Analysis

The pharmacokinetic properties, drug-likeness, and absorption of the compounds under study were evaluated using the SwissADME service [33]. This service was employed to analyze the absorption, distribution, metabolism, and excretion of the compounds. Additionally, BOILED-Egg was used to predict the gastrointestinal absorption and brain penetration of the molecules [44].

## 4. Conclusions

Synthetic compounds **8**–**14** were synthesized with morpholine, pyrrolidine, and piperidine at the 6-position and 4-position of the pyridine and pyrazine rings, respectively, and a thiosemicarbazone chain terminated with piperidine at the 2-position. The first step in the synthesis involved substituting a halogen atom in the starting material, followed by condensation with piperidine-1-carbothiohydrazide via the iminoester formed in situ for **8**–**10** and **14** or directly from the nitrile for **11**–**13**. Compounds **8**–**10** and **12**–**14** complied with the rules for drug candidates, including appropriate bioavailability, absorption in the gastrointestinal tract, and no crossing of the blood–brain barrier. Compounds substituted with basic piperidine (and in one case, pyrrolidine) were the most active (MIC 0.5–4 μg/mL) and the least substituted with morpholine, which lowered the lipophilicity of the molecule. The pyrazine derivatives were also generally less active (MIC even above 512 μg/mL) than the pyridine derivatives. The most promising structures in terms of developing them in the direction of antimycobacterial drugs turned out to be piperidinothiosemicarbazone derivatives with pyrrolidine (**9**) and piperidine (**10**) attached to the pyridine ring in the C-6 position. They exhibited high tuberculostatic activitiy against both standard and resistant strains, with MIC values of 2–4 μg/mL and 0.5–1 μg/mL, respectively. Their potency toward the *M. tuberculosis* Spec. 210 strain was several times higher, compared to the reference drug. The selectivity of action against *M. tuberculosis* of compounds **9** and **10** was demonstrated in extended bioassays and cytotoxicity assays on HaCaT cells.

## Figures and Tables

**Figure 1 pharmaceuticals-16-01267-f001:**
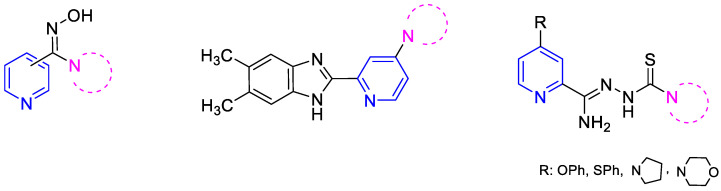
Structural formulas of recently synthesized pyridine derivatives.

**Figure 2 pharmaceuticals-16-01267-f002:**
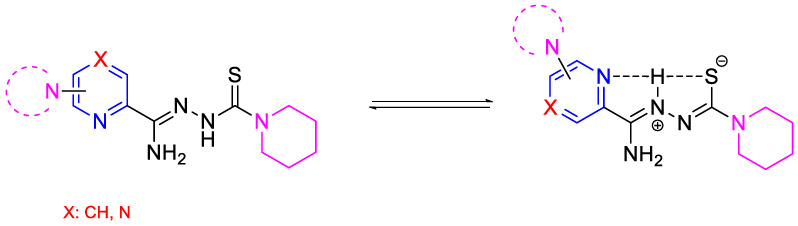
The neutral and possible zwitterionic forms of synthesized and tested compounds.

**Figure 3 pharmaceuticals-16-01267-f003:**
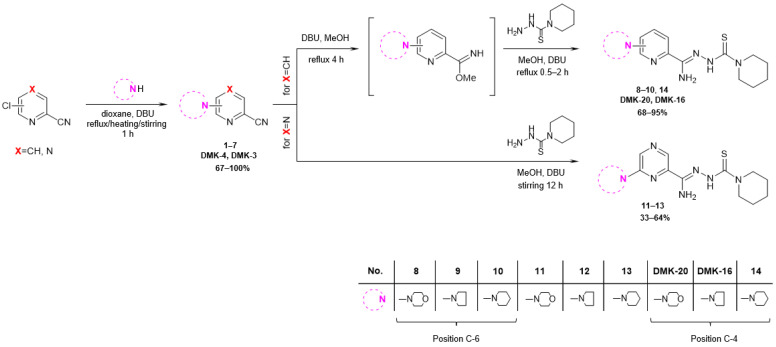
Synthesis pathway of piperidinothiosemicarbazones **8**–**14, DMK-20**, and **DMK-16**.

**Figure 4 pharmaceuticals-16-01267-f004:**
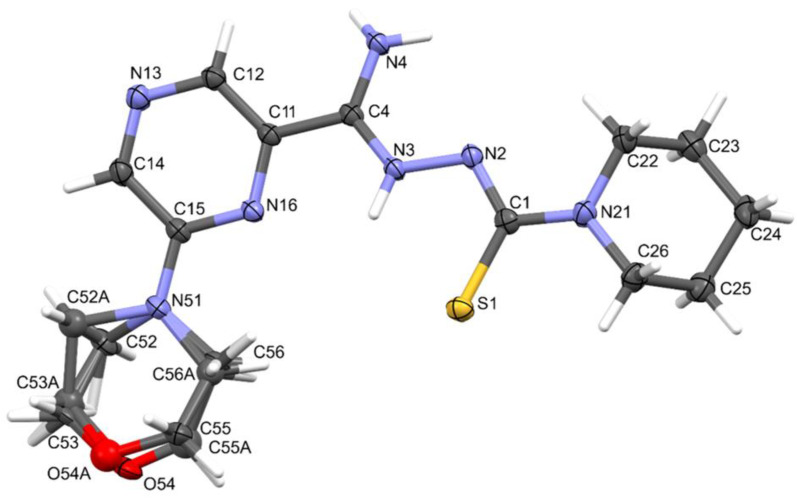
The molecular structure and atom-numbering scheme for compound **11**, with displacement ellipsoids drawn at the 50% probability level.

**Figure 5 pharmaceuticals-16-01267-f005:**
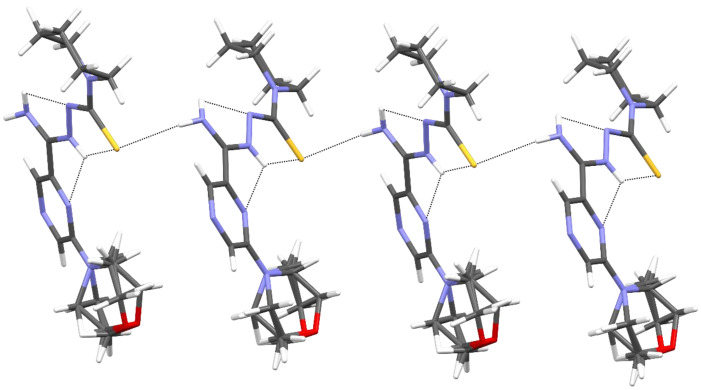
The intermolecular and intramolecular hydrogen bonds in compound **11**.

**Figure 6 pharmaceuticals-16-01267-f006:**
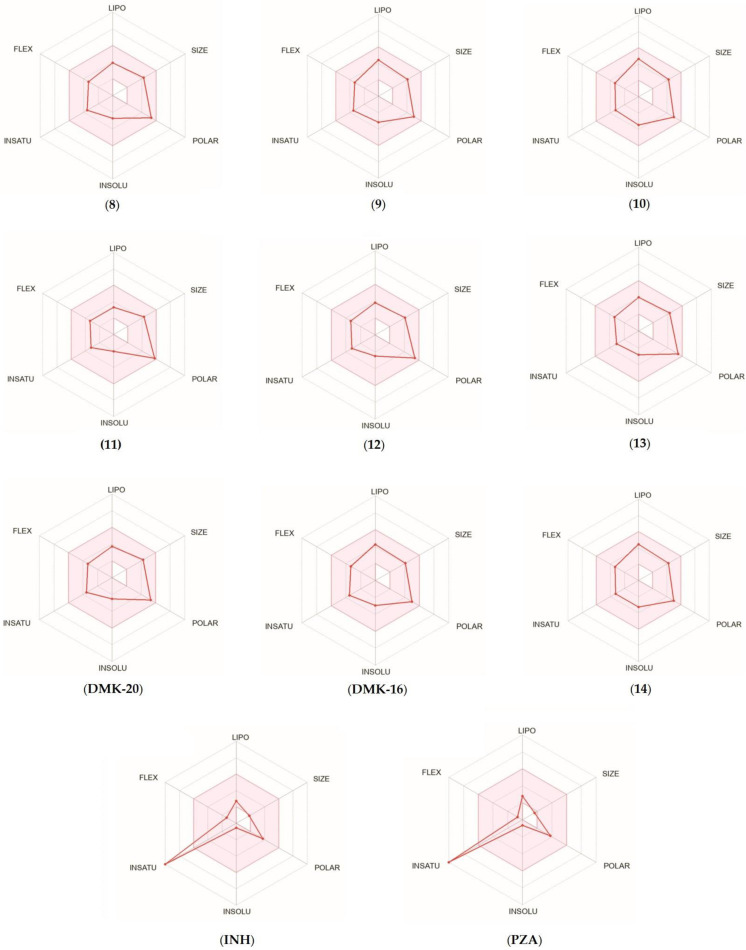
Bioavailability radar for compounds **8**–**14**, **DMK-20**, **DMK-16**, **INH**, and **PZA**.

**Figure 7 pharmaceuticals-16-01267-f007:**
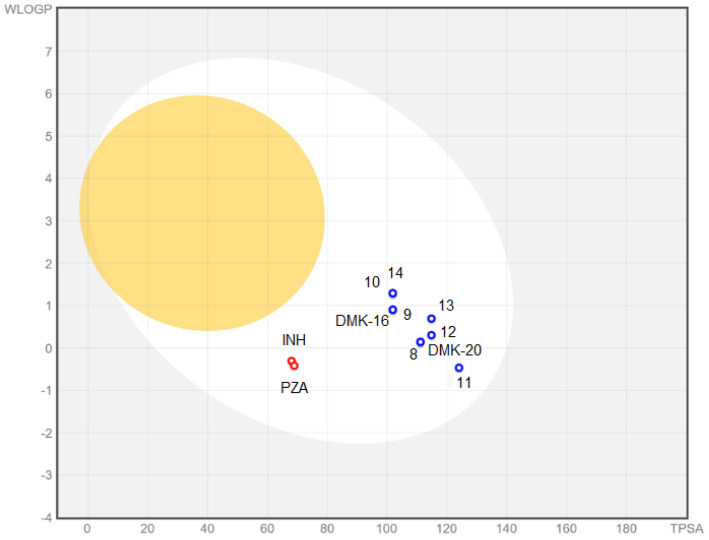
BOILED-Egg plot for compounds **8**–**14**, **DMK-20**, **DMK-16**, **INH**, and **PZA**.

**Table 1 pharmaceuticals-16-01267-t001:** In vitro tuberculostatic activity of compounds **8**–**14**, **DMK-20**, and **DMK-16**.

Compd.	MIC [µg/mL]	
	H_37_Rv	Spec. 210
**8**	16	4
**9**	**4**	**1**
**10**	**2**	**0.5**
**11**	>512	>512
**12**	256	256
**13**	**4**	**4**
**DMK-20**	6.25	6.25
**DMK-16**	6.25	6.25
**14**	**2**	**4**
**INH**	0.125	8

**Table 2 pharmaceuticals-16-01267-t002:** In vitro antimicrobial activity of compounds **9**–**10** and **13**–**14**.

Chemicals Microorganism	9	10	13	14	CIP	VAN	FCZ
	MIC [µg/mL]						
**Gram-positive bacteria**							
*B. cereus* ATCC 10876	>1000	>1000	>1000	0.12	0.12	0.98	-
*B. subtilis* ATCC 6633	>1000	>1000	-	0.12	0.03	0.24	-
*S. epidermidis* ATCC 12228	>1000	>1000	15.6	0.06	0.49	0.98	-
*S. aureus* ATCC 25923	>1000	>1000	1000	0.06	0.49	0.98	-
*M. luteus* ATCC 10240	>1000	>1000	62.5	0.06	0.98	0.12	-
**Gram-negative bacteria**							
*P. aeruginosa* ATCC 9027	>1000	>1000	>1000	500	-	0.49	-
*P. mirabilis* ATCC 12453	>1000	>1000	>1000	250	-	0.03	-
*E. coli* ATCC 25922	>1000	>1000	>1000	500	-	0.004	-
*K. pneumoniae* ATCC 13883	>1000	>1000	>1000	1000	-	0.06	-
**Yeasts**							
*C. parapsilosis* ATCC 22019	1000	>1000	1000	250	-	-	1.95
*C. albicans* ATCC 102231	1000	>1000	>1000	250	-	-	0.98

**Table 3 pharmaceuticals-16-01267-t003:** Cytotoxic activities of compounds **9**–**10**.

Compd.	IC50-HaCaT [µg/mL]		SI IC50-HaCaT/MIC-MT	
	MTT	SULF	MTT	SULF
**9**	>50	>50	>12.5	>12.5
**10**	5.80	>50	2.90	>25

**Table 4 pharmaceuticals-16-01267-t004:** X-ray diffraction data and structure refinement for **11**.

Crystal Data	
Chemical formula	C_7.50_H_11.50_N_3.50_O_0.50_S_0.50_
*M* _r_	174.73
Crystal system, space group	Monoclinic, *P*2_1_/*c*
Temperature (K)	100
*a*, *b*, *c* (Å)	9.7274 (1), 14.3441 (2), 12.6387 (2)
β (°)	96.100 (1)
*V* (Å^3^)	1753.50 (4)
*Z*	8
Radiation type	Cu *Kα*
**Data collection**	
No. of measured, independent, and observed [*I* > 2σ(*I*)] reflections	32,919, 3637, 3482
*R* _int_	0.043
**Refinement**	
*R*[*F*^2^ > 2σ(*F*^2^)], *wR*(*F*^2^), *S*	0.031, 0.078, 1.03
No. of reflections	3637
No. of parameters	244
∆_max_, ∆_min_ (e Å^−3^)	0.28, −0.24

**Table 5 pharmaceuticals-16-01267-t005:** Hydrogen bond geometry (Å, °) for **11**.

D–H⋯A	D–H	H⋯A	D⋯A	D–H⋯A
N3–H3⋯S1	0.88	2.34	2.8330 (10)	116

**Table 6 pharmaceuticals-16-01267-t006:** Calculated values of partition coefficients (log*P*) of compounds **8**–**14**, **DMK-20**, and **DMK-16**.

Compd.	MIC [μg/mL]	log*P*
**8**	16	2.19
**9**	4	2.91
**10**	2	3.32
**11**	>512	0.85
**12**	256	1.57
**13**	4	1.99
**DMK-20**	6.25	1.48
**DMK-16**	6.25	2.19
**14**	2	2.61

## Data Availability

Data is contained within the article and Appendix A.

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
