# Peer review of "Synthesis and Biological Activity of Piperidinothiosemicarbazones Derived from Aminoazinecarbonitriles"

_pharmaceuticals, 2023, doi:10.3390/ph16091267_

Round 1

Reviewer 1 Report (New Reviewer)

D. Ziembicka and co-authors in their article report on the synthesis and biological evaluation of new piperidinothiosemicarbazone derivatives of pyridine and pyrazine series. The aim of this research is to figure out how structural modifications of the previously described compounds would affect their antitubercular and antibacterial activities. For this reason a number of target molecules were synthesized by the reaction of cloro-2-cyanoazines with cyclic amines followed by reactions with piperidine-1-carbothiohydrazide. In addition to biological activity the authors performed X-ray study, quantum-chemical calculations and ADME analysis.

The main achievement of the authors is that they revealed compounds exhibited high tuberculostatic activity towards standard and resistant strains of M. tuberculosis along with low toxicity against non-tumor cells. It seems that antitubercular activity strongly depends on the basicity of amine fragment in azine ring (piperidine>pyrrolidine>morpholine). In this connection it would be of interest to synthesize and test compounds containing more basic moieties such as 1-methylpiperazine or some polycyclic amine fragments in azine ring.

The conclusions are consistent with the evidence and arguments presented.  This manuscript deserves publication in Pharmaceuticals, however, there are some points to be addressed:

1.     Line 359: “…catalytic amount of DBU…” In fact, the amount of DBU is equimolar, not catalytic. As it follows from experimental section the authors usually added 40 mmol of DBU (6 mL) and 40 mmol of the starting compound.

2.     No mass-spectra or HRMS provided in experimental section.

3.     In 1H NMR spectra of all compounds the coupling constants should be given as decimal fractions (3.0 Hz, 8.6 Hz and so on), not whole numbers.

Author Response

  1. Appriopriate correction has been made.
  2. Reports on previously undescribed organic compounds should include, as supplementary data, high-resolution mass spectrometry (HRMS) or elemental analysis. The publication contains an elemental analysis.
  3. Appriopriate correction has been made. The characterization of the tested compounds have been transferred to Supplementary Material.

Reviewer 2 Report (New Reviewer)

The manuscript “Synthesis and Biological Activity of Piperidinothiosemicarbazones Derived from Aminoazinecarbonitriles” by authors Dagmara Ziembicka et al contains the data on synthesis and biological evaluation of thiosemicarbazones derivatives.

Unfortunately, the manuscript has a low level of novelty. It is a set of data without the common idea.

1. Abstract. The first sentence should be deleted.

2. Introduction should be rewritten according to aims and scopes of the manuscript. For example, there are no literature data on antimicrobial activity.

3. The DMK compounds on the Figure 3 are absent. Please add the possible zwitter ionic system onto Figure 3.

4. X-Ray data should be replaced into chemical part. Please provide CCDC number for the structure 11.

4. It is unclear, why authors did QM calculations. The differences between neutral and zwitter ionic form is negligible (0.01-0.03 units). The logP could be pointed from ADME analysis.

5. Authors should provide the supporting information with NMR spectra for the compounds, etc.

Given all the above, I recommend to carry out the essential revision for the manuscript.

Author Response

  1. Appriopriate correction has been made.
  2. Appriopriate correction has been made.
  3. Appriopriate correction has been made. The zwitterionic system was shown on Figure 2.
  4. The CIF files containing full crystallographic data for compound 11 (CCDC 2278736) can be obtained free of charge via http://www.ccdc.cam.ac.uk/conts/retrieving.html. Relevant information on this is provided in Data Availability Statement.
  5. While performing the QM calculations, we wanted to check the charge distribution for both forms and their differences, as this may affect the knowledge of the pharmacophore.
  6. Supplementary Material has been added.

Round 2

Reviewer 2 Report (New Reviewer)

In general, the authors improved the article. I am satisfied with some corrections in the manuscript. Still, I don’t understand why it was necessary to carry out QM calculations. This part have not scientific soundness. These calculations did not explain the difference in antimicrobial activity of compounds. Figure 6 contains a lot of redundant information. I could recommend to delete QM calculations. It would be good if authors add structure-activity relationships analysis. Also, the conclusions should be enhanced. It is need to improve part about biological activity.

Author Response

(1) The QM calculations have been removed from the publication.
(2) Structure-activity relationship analysis is provided in Section 2.5.
(3)
The discussion of the results of biological activity in the conclusions has been improved.

This manuscript is a resubmission of an earlier submission. The following is a list of the peer review reports and author responses from that submission.

Round 1

Reviewer 1 Report

Synthesis and Biological Activity of Piperidinethiosemicarbazones Derived from 6-Aminoazinecarbonitriles from Dagmara Ziembicka et al focuses on the synthesis of 3 new compounds to tested as antituberculous molecules.

The introduction is complete and according to the reported research and the group has experience in this field of research.

Unfortunately, the authors didn't analyse the possibility to synthesized compounds with different cyclic amines which include several and different substituents to explain the potential impact of these differences to the biological results. The authors only used  2 cyclic amines for the pyridine core and only one cyclic amine for the piperidine core. These molecules are not enough to display some relationship between the chemical structure and the biological activity.

I suggest to include a series of derivatives considering several substituents in these cyclic amines to complete the analyses and obtain more precise results regarding the chemical structure impact.

Additionally, it should be noted, as an example, that the cytotoxic activity assay needs to be precise using triplicates and standard deviations information.

Finally, the theoretical analyses (quantum chemical calculations) needs to be more deeply discuss because the wide range of logP and the insufficient structural modifications to compare.

Reviewer 2 Report

The authors discussed the synthesis and biological activity of piperidinethiosemicarbazones derived from 6-aminoazinecarbonitriles. Overall, the manuscript falls short of meeting the publication requirements for 'Pharmaceuticals' due to its lack of novelty and the low quality of all biological results. In addition, there are some specific issues:  

(1) The synthetic procedures lack originality from an organic perspective and have a very limited scope.

(2) While the spectroscopic data for all synthesized compounds are provided, the Supplementary Material is missing.

(3) The main drawback lies in the unsatisfactory results of the antitubercular and antimicrobial activities. I also recommend conducting Docking Studies to complement this work if you will obtain better results with different compounds.

Reviewer 3 Report

The work by Ziembicka et al. describes synthesis and biological activity of three piperidinethiosemicarbazones derived from 6-aminoazinecarbonitriles. The chemistry is very simple and two prepared compounds display significant activity against the resistant Mycobacterium tuberculosis strain spec. 210. The paper is therefore worth of publication subject to minor revision.

 1)      Synthesis of 4 and 5 could be combined to general procedure.

 2)      13C NMR spectra are quoted only at single decimal digit.

 3)      Why the yield of 6 is so low?

 4)      Bioavailability and BOILED-Egg diagram should contain also results for standard(s).

 5)      Table 2 should be omitted as the results are always poor (=/> 1000 ug/mL). One simple sentence discussing results is enough.

 6)      Quantum chemical calculations should be also done (or cited) for standards in order to assess their reliability.

English language should be polished by the native speaker.

E.g. p. 2/line 65 - neither of them = none of them

p. 3/line 93 - initial compounds = starting compounds

p. 5/line 193- 4-6 were subjected to performed to = delete "to performed"

The term "comparator medicine" should be replaced by "standard".

Reviewer 4 Report

Review of the Article  

„ Synthesis and Biological Activity of Piperidinethiosemicarbazones Derived from 6-Aminoazinecarbonitriles” by  Dagmara Ziembicka  , Katarzyna Gobis , Małgorzata Szczesio  , Ewa Augustynowicz-Kopeć  , Agnieszka Głogowska  , Izabela Korona-Głowniak and Krzysztof Bojanowski.

Authors decribed  the synthesies of three new compounds :

 N'-(piperidine-1-carbonothioyl)-6-(pyrrolidin-1-yl)picolinohydrazonamide (4),

 6-(piperidin-1-yl)-N'-(piperidine-1carbonothioyl)picolinohydrazonamide (5)

N'-(piperidine-1-carbonothioyl)-6-(pyrrolidin-1-yl)pyrazine-2-carbohydrazonamide (6)

for testing them as a potential drugs. The synthesis of promissing strutures 4,5 and 6 is described carefully  and preicesly.Particulary  noteworthy is presentation by the Authors the wide spectrum of biological and medicinal tests as : ADME Analysis showing  the optimal range for each property, including lipophilicity , size , polarity , solubility , saturation , and flexibility. The Boiled Egg diagram  is interesting indicated that all compounds showed  absorption in the gastrointestinal tract  and no penetration of the blood–brain barier. Very important is Tuberculostatic Activity Assay,  Antimicrobial Activity Assay, Cytotoxic activity and Quantum Chemical Calculations presenting . calculated values of absolute energy (E),dipole moment (µ), and partition coefficient (logP) in Table 4.

Most of the presented results are promissing for their future application in the medicinal therphy.Unfortunately only compounds 4and 5 are potential  structures for research on an antimycobacterial drug. They exhibited high tuberculostatic activity.

To sum up,the paper presnted for review is an excellent example of how,having recieved „unsatisfactionary” result (see for compound 6) it can be used to deeper to get to  know,better understand and explain its lower than expected biological activity.